# Dietary Fiber Inulin Improves Murine Imiquimod-Induced Psoriasis-like Dermatitis

**DOI:** 10.3390/ijms241814197

**Published:** 2023-09-17

**Authors:** Mai Yoshida, Yoko Funasaka, Hidehisa Saeki, Masami Yamamoto, Naoko Kanda

**Affiliations:** 1Department of Dermatology, Nippon Medical School, Bunkyo City 113-8602, Tokyo, Japan; m-mai@nms.ac.jp (M.Y.); funasaka@nms.ac.jp (Y.F.); h-saeki@nms.ac.jp (H.S.); 2Department of Applied Science, School of Veterinary Nursing and Technology, Nippon Veterinary and Life Science University, Musashino 180-8602, Tokyo, Japan; masami@nvlu.ac.jp; 3Department of Dermatology, Nippon Medical School Chiba Hokusoh Hospital, Inzai 270-1694, Chiba, Japan

**Keywords:** psoriasis, high-fiber diet, inulin, imiquimod, propionate, interleukin-17, gut microbiome, *Bacteroides*

## Abstract

Psoriasis is a chronic skin disease with interleukin (IL)-17-dominated inflammation and hyperproliferation of epidermis. Dietary fiber is fermented by the gut microbiome into short-chain fatty acids (SCFAs) that manifest anti-inflammatory effects. We examined if feeding with an inulin-enriched high-fiber diet (HFD) might improve topical imiquimod-induced psoriasis-like dermatitis in mice. HFD reduced thickening and total severity scores of imiquimod-induced dermatitis and reduced epidermal thickness, inflammatory infiltrates, including Ly6G+ neutrophils, and epidermal Ki67+ proliferating cells. HFD reduced mRNA levels of IL-17A, IL-17F, IL-22, IL-1β, tumor necrosis factor (TNF)-α, CXCL1, CXCL2, and keratin 16 and increased those of transforming growth factor (TGF)-β1 and cyclin-dependent kinase inhibitor 1A in imiquimod-induced dermatitis. In 16S rRNA sequencing of the gut microbiome, imiquimod increased relative abundance of phylum Firmicutes, while HFD increased that of phylum Bacteroidota and genus *Bacteroides*. HFD increased serum and fecal concentrations of SCFA propionate. Oral propionate reduced inflammatory infiltrates and epidermal Ki67+ cells and reduced mRNA levels of IL-17A, IL-17F, IL-17C, IL-22, IL-1β, IL-6, TNF-α, CXCL1, CCL20 and increased those of TGF-β1and IL-10 in imiquimod-indued dermatitis. Dietary inulin supplementation improves imiquimod-induced psoriasis-like dermatitis partially via propionate, and may be a promising adjunctive therapy for psoriasis.

## 1. Introduction

Psoriasis is a recurrent inflammatory keratosis manifesting scales, erythema, and skin thickness. Psoriasis features an enhanced tumor necrosis factor (TNF)-α–interleukin-23 (IL-23)–IL-17 immune axis and exaggerated proliferation and abnormal differentiation of epidermal keratinocytes [1]. Dendritic cells (DCs) triggered by certain stimuli secrete TNF-α, which stimulates themselves and promotes their IL-23 secretion. IL-23 induces type 17 helper T (Th17) cells to produce IL-17A and IL-22, which act on keratinocytes to induce their proliferation and secretion of cytokines or chemokines that attract neutrophils, monocytes, and lymphocytes. Innate immune cells, such as type 3 innate lymphoid cells, γδT cells, or mucosal-associated invariant T cells, also secrete IL-17A, and are involved in the development of psoriasis [2,3]. Regulatory T cells (Tregs) inhibit the proliferation of effector T cells and sustain homeostasis and self-tolerance. Tregs of psoriasis patients are numerically and/or functionally inferior compared to healthy subjects, and cannot sufficiently suppress the activity of Th17 cells [4,5,6].

Various genetic or environmental factors are related to the development of psoriasis [2]. The environmental factors include dietary habits. The modification of diet changes the composition of the gut microbiota. Previous studies showed that Western diet characterized by high fat, high sugar, high salt, and low fiber induced gut dysbiosis and exaggerated psoriasis-like dermatitis in mice [5]. Patients with psoriasis are often associated with gut dysbiosis [6]. The gut dysbiosis may alter the profiles of microbial metabolites, induce intestinal inflammation, enhance gut permeability, and trigger inflammation in distant organs. Conversely, the supplementation of dietary fiber shows anti-inflammatory effects [7]. Dietary fiber represents carbohydrates that are indigestible in the small intestine, but can be fermented by microbiome in colon to generate short-chain fatty acids (SCFAs), such as acetate, propionate, or butyrate. SCFAs are absorbed in colon epithelial cells or transported into the circulation to reach distant tissues including skin. SCFAs are reported to promote the generation of Tregs via inducing the expression of forkhead box P3 (Foxp3) [8] or to regulate the production of proinflammatory cytokines/chemokines by modulating signaling pathways such as NF-κB [9], p38 mitogen-activated protein kinase [10], mechanistic target of rapamycin (mTOR) [9] via suppression of histone deacetylase (HDAC) [11,12] or through G-protein-coupled receptors, GPR41, GPR43, or GPR109A [13]. SCFAs manifest both anti- and pro-inflammatory effects dependent on the concentration and immunological milieu [14,15]. The supplementation of dietary fiber also changes the composition of the gut microbiome and stimulates the growth of beneficial bacteria, such as *Bifidobacterium*, leading to the regulation of intestinal and systemic inflammation. Previous studies report that the supplementation of dietary fiber attenuates chronic inflammatory diseases in the colon [16], joints [17,18], central nervous system [19], or airway [20] in mice or humans. However, there are very few studies regarding its effects on psoriasis [21].

We herein investigated if a high-fiber diet (HFD) enriched with inulin might attenuate psoriasis-like dermatitis induced by topical application of imiquimod, a Toll-like receptor (TLR)7 agonist in mice. Inulin is fructose polymers, consisting of 2 to 60 fructose units connected by β (2,1) bonds and chain-terminating glucosyl moieties, and is derived from plants such as wheat, onion, bananas, garlic, asparagus, or chicory. Human psoriasis lesions and imiquimod-induced psoriasiform dermatitis manifest the enhanced expression of IL-17-related inflammatory or proliferative markers, such as IL-17A/C/F, IL-22, TNF-α, IL-1β, IL-6, keratin 16, and the reduced expression of regulatory or anti-proliferative markers, such as IL-10, transforming growth factor (TGF)-β1, cyclin-dependent kinase inhibitor (CDKN) 1A [1], and the abundant infiltration of CD3+ T cells and Ly6G+ neutrophils and increase in Ki67+ proliferating epidermal keratinocytes [22], and are associated with the defects of Foxp3+ Tregs [4]. We herein investigated if the inulin supplementation might reverse the dysregulated expression of the above markers in imiquimod-induced psoriasiform dermatitis. We have obtained the results that the HFD attenuated imiquimod-induced dermatitis together with an increase in systemic levels of SCFA propionate, and oral propionate partially attenuated the dermatitis, indicating that an HFD-induced increase in systemic propionate derived from dietary inulin might mediate the attenuation of the dermatitis.

## 2. Results

### 2.1. HFD Improved Murine Imiquimod-Induced Psoriasis-like Dermatitis

#### 2.1.1. HFD Reduced the Thickening and Total Scores of Imiquimod-Induced Dermatitis

Topical imiquimod on shaved back skin induced psoriasis-like eruption with skin thickening, erythema, and scales (Figure 1b,d–g, red lines), while these features were not revealed in Vaseline-treated mice (Figure 1a,d–g, light-blue lines). Feeding with inulin-enriched HFD greatly reduced thickening scores and partially but significantly reduced total and scaling scores (Figure 1c–g, purple lines).

#### 2.1.2. HFD Reduced Acanthosis and Number of Infiltrating Inflammatory Cells

The imiquimod-treated skin revealed psoriasis-like histology showing epidermal thickness with elongation of rete ridges, hyperkeratosis, parakeratosis, and abundant inflammatory infiltrates in the dermis (Figure 2b,d,e, middle bars). HFD reduced the epidermal thickness and number of inflammatory cells infiltrating into imiquimod-induced dermatitis (Figure 2c–e, right bars).

Topical imiquimod increased the number of epidermal Ki67+ proliferating cells, which were reduced by the HFD (Figure 3a–d). Topical imiquimod increased the number of infiltrating Ly6G+ neutrophils, which were reduced by the HFD (Figure 3e–h). Topical imiquimod increased the number of CD3+ T cells, while the HFD appeared to decrease the cell number; however, the difference was not significant (Figure 3i–l). The number of Foxp3+ cells was very small in ND-fed, Vaseline-treated mice (NDV), and was slightly increased in ND-fed, imiquimod-treated mice (NDI) and HFD-fed, imiquimod-treated mice (HFDI) without a significant difference between NDI versus HFDI (Figure 3m–p). Those findings suggest that the HFD may reduce the number of proliferating keratinocytes and infiltrating neutrophils in imiquimod-induced dermatitis.

#### 2.1.3. HFD Decreased mRNA Expression of Inflammatory Cytokines/Chemokines and Keratin 16 and Increased That of TGF-β1 and CDKN1A

We analyzed the effects of the HFD on mRNA expression of pro-inflammatory cytokines/chemokines or proliferation-related molecules in imiquimod-induced dermatitis by quantitative real-time polymerase chain reaction (qPCR). The HFD reduced mRNA expression of IL-17A, IL-17F, IL-22, TNF-α, IL-1β, CXCL1, and CXCL2 induced by imiquimod (Figure 4). The HFD appeared to reduce mRNA expression of IL-17C, IL-6, and CCL20 induced by imiquimod; however, the differences between NDI and HFDI were not significant. The HFD significantly increased mRNA expression of anti-inflammatory cytokine TGF-β1 in imiquimod-induced dermatitis, and appeared to augment that of another anti-inflammatory cytokine IL-10, though the difference between NDI and HFDI was not significant. The HFD decreased mRNA expression of hyperproliferation-associated keratin 16 induced by imiquimod, and increased that of cell cycle inhibitor CDKN1A in imiquimod-induced dermatitis. The HFD did not alter mRNA expression of interferon (IFN)-γ or Foxp3. These results indicate that the HFD suppressed mRNA expression of pro-inflammatory cytokines/chemokines and proliferation-associated keratin 16 and increased that of anti-inflammatory cytokine TGF-β1 and cell cycle inhibitor CDKN1A in imiquimod-induced dermatitis, and these effects might be related to the suppression of epidermal hyperproliferation, inflammatory infiltration, and severity of dermatitis.

### 2.2. Effects of Imiquimod and HFD on the Composition of the Gut Microbiome

We then examined the composition of the gut microbiome in NDV, NDI, and HFDI by 16S rRNA sequencing of fecal samples. Regarding α diversity, the Chao 1 index (Figure 5a), reflecting richness of the species, of NDI is significantly higher than that of NDV, while that of the HFDI is significantly lower than those of NDV and NDI. The Simpson index (Figure 5b), reflecting the evenness of individual species, of NDI is not different from that of NDV, and that of the HFDI appeared lower than those of NDV and NDI, but the differences were not significant. The Jaccard β diversity (Figure 5c), reflecting number of common species, showed clear segregation among NDV, NDI, and HFDI.

At phylum levels, NDI increased relative abundance of *Firmicutes* and reduced that of *Bacteroidota* (synonym of Bacteroidetes) compared with NDV (Figure 5d). HFDI showed highly increased abundance of phylum *Bacteroidota* and reduction in *Firmucutes* compared with NDI (Figure 5d). The ratio of *Bacteroidota*/*Firmicutes* in HFDI was significantly higher than those of NDV and NDI (Figure 5e).

At genus levels, the relative abundance of genus *Bacteroides* in NDI appeared lower than that in NDV though the difference was not statistically significant while that in HFDI is remarkably higher than those in NDV and NDI with significant differences (Figure 6a,b).

Linear discriminant analysis (LDA) and effect size (LEfSe) clarified taxa with the greatest differences in abundance at phylum, class, order, family and genus levels between individual groups. Between NDI versus NDV (Figure 7a), NDI augmented the abundance of Firmicutes and Desulfobacterota, but decreased that of Bacteroidota and Actinobacteriota at phylum levels relative to NDV. At genus or family levels, NDI increased the abundance of genera *Lachnospiraceae_UCG-001*, *A2*, *NK4A136*, and *FCS020_group*, *Oscillospirales UCG-010*, *Staphylococcus*, *Turicibacter*, *Desulfovibrio*, *Corynebacterium* and reduced genera *Muribaculaceae*, *Enterorhabdus*, *Gemella*, and *Erysipelatoclostridium* relative to NDV.

Between HFDI versus NDI (Figure 7b), HFDI augmented the abundance of Bacteroidota and Actinobacteriota and decreased that of Firmicutes and Desulfobacterota relative to NDI at phylum levels. At genus or family levels, HFDI increased genera *Bacteroides*, *Bifidobacterium*, *Faecalibaculum*, *Candidatus_Stoquefichus*, *Clostridium_innocuum_group*, *Marvinbryantia* and family *Enterobacteriaceae* and reduced genera *Lactococcus*, *Blautia*, *Peptococcus*, *Lachnospiraceae_UCG-001*, *Clostridia_vadinBB60_group*, *Staphylococcus*, *Streptococcus*, *Desulfovibrio*, *Helicobacter*, *Muribaculaceae*, and *Corynebacterium* relative to NDI.

Overall, imiquimod increased the abundance of phylum Firmicutes and reduced the abundance of phylum Bacteroidota. The HFD augmented the abundance of phylum Bacteroidota, especially that of genus *Bacteroides* in this phylum, and reduced the abundance of phylum Firmicutes.

### 2.3. HFD Increased Serum and Fecal Levels of Propionate

Since dietary inulin is fermented to generate SCFAs by the gut microbiome, we analyzed the impact of the HFD on serum (Figure 8a–d) and fecal levels of SCFAs (Figure 8e–h). Both serum (Figure 8b) and fecal (Figure 8f) propionate levels of the HFDI were significantly higher than those of NDI and NDV. Fecal acetate level of the HFDI was significantly lower than those of NDI and NDV (Figure 8e), while there were no differences in serum levels of acetate among the three groups (Figure 8a). There were no differences in serum and fecal butyrate (Figure 8c,g) or β-hydroxyl-butyrate levels (Figure 8d,h) among the three groups. The concentrations of these SCFAs in skin lesions of NDV, NDI, and HFDI were less than the limit of quantitation. These results suggest that the HFD might increase the systemic levels of propionate, and the effect might be generated via the fermentation of inulin by the gut microbiome.

### 2.4. Oral Propionate Partially Attenuated Imiquimod-Induced Psoriasis-like Dermatitis

#### 2.4.1. The Impact of Oral Propionate on Clinical and Histological Findings in Imiquimod-Induced Dermatitis

HFD-induced increase in systemic propionate levels indicate that propionate derived from dietary inulin might mediate the HFD-induced attenuation of imiquimod-induced dermatitis. We thus examined the effects of oral propionate on the imiquimod-induced dermatitis. Mice were fed with ND plus sodium propionate or control saline in drinking water, then imiquimod was topically applied, and the clinical and histological findings were compared. Thickening scores of ND-fed, imiquimod-treated, propionate-drinking mice (NDIP) (Figure 9c,e, purple line) appeared modestly reduced compared to ND-fed, imiquimod-treated, saline-drinking mice (NDIS) (Figure 9b,e, red line); however, the differences were not significant. Total severity scores of NDIP (Figure 9g, purple line) also appeared slightly reduced compared to NDIS (Figure 9g, red line); however, the differences were not significant.

In histology, the number of infiltrating cells in NDIP (Figure 10c,e, right bar) was significantly lower than that in NDIS (Figure 10b,e, middle bar), while the epidermal thickness in NDIP (Figure 10d, right bar) was not significantly different from that in NDIS (Figure 10d, middle bar).

In immunohistochemistry, the number of epidermal Ki67+ proliferating cells in NDIP was significantly lower than that in NDIS (Figure 11a–d). The numbers of Ly6G+ neutrophils (Figure 11e–h) and CD3+ T cells (Figure 11i–l) in NDIP were significantly lower than those in NDIS. No significant differences existed in the number of Foxp3+ cells among ND-fed, Vaseline-treated, and saline-drinking mice (NDVS), NDIS, and NDIP (Figure 11m–p). These results indicate that oral propionate reduced the numbers of infiltrating neutrophils and T cells, and of proliferating epidermal keratinocytes in imiquimod-induced dermatitis.

#### 2.4.2. The Impact of Oral Propionate on mRNA Expression of Pro- or Anti-Inflammatory Cytokines/Chemokines or Proliferation-Related Molecules

Oral propionate decreased mRNA expression of IL-17A, IL-17C, IL-17F, IL-22, IL-6, IL-1β, TNF-α, CXCL1, and CCL20 in imiquimod-induced dermatitis in comparison between NDIS versus NDIP (Figure 12). Oral propionate increased mRNA expression of TGF-β1 and IL-10 in imiquimod-induced dermatitis. The mRNA level of CDKN1A in NDIP appeared higher than that in NDIS; however, the difference was not significant. The mRNA level of CXCL2 in NDIP appeared lower than that in NDIS; however, the difference was not significant. The mRNA levels of keratin 16, IFN-γ, and Foxp3 in NDIP were not significantly different from those in NDIS.

## 3. Discussion

This study has proven that inulin-enriched HFD might be useful for inhibiting imiquimod-induced psoriasis-like dermatitis via induction of propionate. In this study, inulin-enriched HFD augmented genus *Bacteroides* in the gut microbiome, which is consistent with previous studies using inulin [23]. Serum and fecal propionate, increased by the HFD, might be the fermentation product of inulin by the gut microbiome, and may be absorbed into the hepatic portal vein, then into systemic circulation, carried to peripheral organs. Genus *Bacteroides* is reported to abundantly produce propionate [24,25], and may be a major bacterium producing propionate in HFD-fed mice. It is also reported that oral administration of propionate-producing *Bacteroides* species attenuated complete Freund’s adjuvant-induced arthritis in mice [26]. High-fiber diet-induced increase in genus *Bacteroides* abundantly producing propionate may contribute to the increased systemic levels of propionate and resultant attenuation of imiquimod-induced psoriasiform dermatitis by propionate. The HFD also increased relative abundance of genus *Bifidobacterium*, and similar results are reported in previous studies using inulin [27]. Genus *Bifidobacterium* produces propionate [28], and propionate accelerates the growth of *Bifidobacterium* [29]. Genus *Bifidobacterium* may thus be another propionate producer in HFD-fed mice.

Topical imiquimod reduced phylum Bacteroidota and increased that of Firmicutes compared to Vaseline-treated control mice, which is similar to the reported gut microbial composition in psoriasis patients compared to healthy individuals [30,31]. Meanwhile, both lower [6] and higher [32] α diversity of the gut microbiome are reported in psoriasis patients relative to healthy individuals. In this study, α diversity was slightly increased by imiquimod and greatly reduced by the HFD. It is reported that propionate suppresses the growth of Gram-negative facultative and obligatory anaerobes such as *Enterobacter cloacae* and *Escherichia coli* [29]. The HFD-induced reduction in α diversity might thus be at least partially caused by propionate possibly produced by genus *Bacteroides* or *Bifidobacterium*. It is also reported that inulin-enriched feeding in CD1 nude mice reduced the Shanon index of the gut microbiome in association with the increased abundance of *Bacteroides acidifaciens* [23]. Previous studies reported that inulin intake increased species of the *Bacteroides fragilis* group, such as *Bacteroides acidifaciens*, in the gut microbiome [23,33,34]. We should further identify the species among genus *Bacteroides* increased by the HFD in our model mice.

The improvement in imiquimod-induced dermatitis by the HFD may be at least partially mediated by propionate, since most of the effects of the HFD were reproduced by oral sodium propionate. It is known that propionate regulates gene expression via inhibition of HDACs [35,36]. HDACs are enzymes that delete acetyl moieties from lysine residues at histone tails, which strengthens the interaction between histones and DNA, making compact chromatin and inhibiting the recruitment of transcriptional apparatus to the promoter, leading to the repression of gene transcription. HDACs also deacetylate non-histone proteins such as transcription factors, signal mediators, or chaperones. Conversely, HDAC inhibitors induce hyperacetylation of histones and non-histone proteins; the former induces the transcription of epigenetically repressed genes and the latter up- or down-regulates the activities of transcription factors, signal mediators, or chaperones [11,12]. HDAC inhibitors suppress NF-κB-mediated transcription of genes encoding proinflammatory cytokines/chemokines by several mechanisms: HDAC inhibitors increase acetylation of NF-κB p65, which reduces its nuclear translocation and DNA binding [37], suppress proteasomal activity to degrade IκBα [38], or inhibit the access of RNA polymerase II to NF-κB-targeting genes [39]. Oral propionate and the HFD reduced mRNA expression of IL-17A, IL-17C, IL-17F, IL-22, IL-1β, IL-6, TNF-α, CXCL1, CXCL2, and CCL20 in imiquimod-induced dermatitis. Since NF-κB can induce the transcription of these cytokine/chemokine genes, the suppression of NF-κB activity might be involved in their reduced expression by propionate and HFD.

In imiquimod-induced dermatitis, IL-17A, IL-17F, and IL-22 may be mostly derived from Th17 cells or γδT17 cells, while CXCL1, CXCL2, CCL20, and IL-17C may be mainly produced by epidermal keratinocytes. IL-6, TNF-α, and IL-1β may be produced by various types of cells, such as T cells, DCs, macrophages, keratinocytes, or fibroblasts. The target cell types for the propionate- or HFD-induced reduction in individual cytokines/chemokines should further be identified. The HFD- or propionate-induced reduction in CXCL1/2 or CCL20 expression might result in the reduced infiltration of neutrophils or T cells/γδT cells in the imiquimod-induced dermatitis, respectively.

The HFD and propionate increased the expression of the anti-inflammatory cytokine TGFβ1. It is reported that SCFAs promote TGF-β1 expression through transcription factor specificity protein 1 (Sp1) in human intestinal epithelial cells [40]. HDAC inhibitors induce Sp1-dependent gene expression by histone acetylation of gene promoters, which facilitates the binding of Sp1 [41] or the recruitment of transcriptional activator cyclic AMP response element-binding protein-binding protein [42,43]. A similar mechanism is considered for the propionate- or HFD-induced expression of TGF-β1. TGF-β1 regulates inflammatory T cell responses: it limits the response to IL-23 in Th17 cells by suppressing the expression of IL-23 receptor [44]. TGF-β1 is ubiquitously expressed in keratinocytes, fibroblasts, DCs, macrophages, or T cells, and target cell types and mechanisms for propionate- or HFD-induced TGF-β1 expression should further be identified.

Oral propionate greatly and the HFD modestly increased the expression of IL-10 in imiquimod-induced dermatitis. It is reported that propionate augmented IL-10 expression in murine naïve T cells via HDAC inhibition-induced acetylation of mTOR-p60 S6 kinase and resultant phosphorylation of signal transducer and activator of transcription (STAT) 3 [45] or in murine Th1 cells by inducing the expression of transcription factor B lymphocyte-induced maturation protein 1 through the mTOR/STAT3 pathway via GPR43 [46]. Similar mechanisms may mediate propionate- or HFD-induced IL-10 expression. Since the other types of cells different from T cells, such as DCs, macrophages, keratinocytes, or fibroblasts, can produce IL-10, target cell types and precise mechanisms for propionate- or HFD-induced IL-10 expression should further be elucidated.

HFD and oral propionate did not increase Foxp3 mRNA levels or number of Foxp3+ cells in imiquimod-induced dermatitis. It is hypothesized that propionate and HFD may not induce the generation of Foxp3+ Tregs in imiquimod-induced dermatitis, which might contain a large amount of IL-6, IL-1β, or IL-23 counteracting their generation. A previous study also reported that oral propionate did not induce Foxp3+ Tregs though significantly attenuated intestinal inflammation in HLAB27/β2-microglobulin-transgenic rats [47]. It is reported that SCFAs enhance the induction of Foxp3+ T cells only under conditions of low T-cell activation, such as low levels of CD3 activation [15].

The HFD greatly reduced the epidermal thickness clinically and histologically together with decreasing number of Ki67+ proliferating keratinocytes and mRNA level of keratin 16 and increasing mRNA level of CDKN1A, which induces cell cycle arrest. The results indicate that the HFD might suppress epidermal hyperproliferation through downregulating factors for cell proliferation and promoting factors for cell cycle arrest. Meanwhile, oral propionate significantly decreased the number of epidermal Ki67+ proliferating keratinocytes, and modestly decreased thickening score and increased *cdkn1a* expression without statistical significance. The results are analogous to a recent study showing that dietary inulin prevents the growth of estrogen receptor-negative mammary carcinoma in Her2/neu transgenic mice by HDAC inhibition partially via propionate whose plasma levels are increased by inulin [48]. It is reported that propionate in vitro suppresses the growth of epithelial HaCaT cells [49], or of breast, lung or colon carcinoma by promoting cell cycle arrest in the G1 phase [50,51,52,53]. It is also reported that HDAC inhibitors induce *cdkn1a* gene transcription via histone acetylation of chromatin [54]. It is thus indicated that the increased expression of *cdkn1a* by the HFD might be at least partially mediated via HDAC inhibition by propionate derived from inulin.

Compared to the prominent inhibition of epidermal hyperproliferation by the HFD, the anti-proliferative effects of oral propionate were modest without inhibiting keratin 16 expression. The difference may be possibly because the HFD may induce other metabolites different from SCFAs via alteration of the gut microbiome, and such putative metabolites might suppress hyperproliferation of keratinocytes. One possible candidate is bile acids: an inulin-rich diet is reported to augment systemic levels of bile acids via promoting cecal excretion of bile acids and their reabsorption into circulation [27,55]. Genera *Bacteroides* and *Bifidobacterium*, whose relative abundance is increased by the HFD, have enzymes, bacterial salt hydrolases that hydrolyze conjugated bile acids into unconjugated ones and modulate bile acid metabolism [56]. Oral administration of bile acids, lithocholic acid (LCA), deoxycholic acid, and 3-oxoLCA, reduced epidermal thickness and number of epidermal Ki67+ cells in psoriasis-like dermatitis of IL-23 minicircle DNA-delivered mice [57]. Deoxycholate, chenodeoxycholate, and LCA reduced the proliferation of human keratinocytes in vitro [58]. Oral administration of *Bifidobacterium breve* CCFM683 reduced keratin 16 expression in imiquimod-induced dermatitis with increased concentrations of several bile acids in the colon, indicating involvement of certain bile acids in down-regulation of keratin 16 expression [59]. Another candidate is polysaccharide A, a cell wall component of *Bacteroides*. Polysaccharide A is known to suppress the growth of colorectal carcinoma in vitro with reducing transcription of cell cycle inducer genes, *ccnd1* and *cdk2* and increasing transcription of cell cycle inhibitor gene *cdkn1b* [60]. The involvement of these or other unknown metabolites in the anti-proliferative effects of the HFD should further be examined.

This study has some limitations. Firstly, we used only imiquimod-induced dermatitis as a psoriasis mouse model. The effects of the HFD should further be evaluated in the other psoriasis models, such as human non-lesional psoriasis skin-transplanted model (SCID-hu), global gene-manipulated (B27/h*β*_2_m-transgenic rat, *IL1rn*−/− mice), keratinocyte-specific gene-manipulated (K14-VEGF, K5.Stat3C), intradermal IL-23-injected or IL-23 mini-circle DNA-delivered mice, or autoantigen-recognizing T cell-transferred Rag−/− mice (Dsg3H1) [22], or UVB photodermatitis rats [61]. Secondly, the other mediators different from propionate might be involved in HFD-induced attenuation of imiquimod-induced dermatitis. The possible mediators should further be identified. Thirdly, the causal relationship between HFD-induced alteration of the gut microbiome and attenuation of dermatitis is uncertain and should be clarified by further studies using oral gavage administration of genus *Bacteroides*. Fourthly, the signaling molecules involved in HFD- or propionate-induced attenuation of psoriasis-like dermatitis have not been clarified and should comprehensively be analyzed, such as NF-κB, STAT3, mTOR, or TLR7. Lastly, we investigated if supplementation of inulin in a normal diet, not a Western diet or high-fat diet, may attenuate the imiquimod-induced psoriasis-like dermatitis. However, it is reported that a Western diet containing high sugar and high fat exacerbated imiquimod-induced psoriasis-like dermatitis [62] and topical imiquimod treatment induced hyperglycemia via impairment of insulin secretion in mice [63]. Further study should examine if inulin supplementation in a Western diet may counteract the Western diet-induced exacerbation of imiquimod-induced dermatitis or imiquimod-induced hyperglycemia.

In conclusion, inulin-enriched HFD reduced thickening and total scores of imiquimod-induced psoriasiform dermatitis and histologically reduced epidermal thickness and inflammatory infiltrates, including Ly6G+ neutrophils. The HFD decreased mRNA expression of IL-17A, IL-17F, IL-22, TNF-α, IL-1β, CXCL1, CXCL2, and keratin 16, and increased mRNA expression of TGF-β1 and CDKN1A in imiquimod-induced dermatitis. The HFD increased the abundance of phylum Bacteroidota and genus *Bacteroides* in the gut microbiome and serum and fecal propionate levels. The effects of the HFD in attenuating imiquimod-induced dermatitis may be at least partially mediated by propionate. These results indicate that dietary inulin supplementation may be a promising adjunctive therapy for psoriasis complementing current pharmacological therapies. Further research should elucidate the precise mechanisms for inulin-induced attenuation of psoriasis-like dermatitis.

## 4. Materials and Methods

### 4.1. Mice

BALB/c mice were obtained from CLEA Japan, Inc. (Tokyo, Japan). All mice for experiments were 6- to 8-week-old females and were maintained in specific pathogen-free conditions at the animal facility of Nippon Medical School.

### 4.2. Reagents

Rabbit monoclonal anti-mouse CD3 (SP7) was obtained from Abcam (Cambridge, UK). Rat monoclonal anti-mouse Ly6G (1A8) was obtained from Aviva Systems Biology, Corp. (San Diego, CA, USA). Rat monoclonal anti-mouse Foxp3 and rabbit polyclonal anti-Ki-67 were obtained from Thermo Fisher Scientific (Waltham, MA, USA). Sodium propionate was obtained from Wako Pure Chemical Industries, Ltd. (Osaka, Japan).

### 4.3. Development of Imiquimod-Induced Psoriasis-like Dermatitis

Mice were fed with either an HFD (D14071803, 10% inulin; Research Diets, New Brunswick, NJ, USA) or ND (D14071802, 10% cellulose; Research Diets) for 3 weeks, then 45 mg of 5% imiquimod cream (Beselna cream; Mochida Pharmaceuticals, Tokyo, Japan) or Vaseline (control) was applied to the shaved back skin for 5 consecutive days (days 0–4). A blinded investigator performed daily evaluations for clinical severity scoring based on the psoriasis area and severity index. Erythema, scaling, and thickening were each scored independently on a scale from 0 to 4 (0, none; 1, slight; 2, moderate; 3, marked; 4, very marked), and the sum of the scores was used as the total clinical score (scale 0–12) [23]. Each group consisted of three mice, and a series of experiments was carried out 6 times. In another series of experiments, mice were fed with ND and 200 mM sodium propionate or normal saline in drinking water for 3 weeks, then treated with imiquimod or Vaseline as above.

### 4.4. Histological Analyses

At day 5, murine back skin samples were obtained, and one half of the sample was paraffin-embedded for hematoxylin–eosin staining while the other half was embedded in Optimal Cutting Temperature Compound (Sakura Finetek Japan, Tokyo, Japan), and snap-frozen in liquid nitrogen for immunohistochemical staining. The frozen samples were cut into 5 μm-thick sections, fixed with cold acetone, and washed with phosphate-buffered saline. Samples were incubated with blocking solution (3.75% bovine serum albumin/5% goat serum, Zymed, Carlsbad, CA, USA) for 30 min, and incubated overnight at 4 °C with the primary antibodies (anti-CD3, Ly6G, Ki67, or Foxp3 antibodies) (Table 1) or isotype controls in appropriate dilutions. After washing, slides were incubated with the secondary antibodies for 2 h. The slides were washed again and incubated with an avidin–biotin peroxidase complex followed by Nova RED (Vector Labs, Burlingame, CA, USA) and counterstained with Mayer’s hematoxylin. Epidermal thickness was measured, and numbers of inflammatory infiltrating cells, Ki67+, CD3+, Ly6G+, or Foxp3+ cells, were counted under ×400 high-power fields in five random grids per section.

### 4.5. qPCR

Total RNA was isolated from murine back skin samples at day 5 using Nucleospin RNA (Takara Bio Inc, Tokyo, Japan), and cDNA was synthesized using a High-Capacity RNA-to-cDNA™ Kit (Thermo Fisher Scientific, Waltham, MA, USA). Gene expression of each sample was quantified in duplicate using the TaqMan™ Fast Advanced Master Mix (Thermo Fisher Scientific). Primers were purchased from Thermo Fisher Scientific (Table 1). Gene expression levels were normalized to those of the GAPDH gene as an internal control. The relative expression levels of each gene were determined by the 2^−ΔΔCT^ method.

### 4.6. Measurement of the Concentrations of SCFAs in Feces and Sera

At day 5, fresh feces were obtained, and blood was collected by cardiopuncture from NDV, NDI, and HFDI mice (4 mice per group). The feces and sera were stored at −80 °C until analysis. The concentrations of SCFAs were measured by laboratory testing service (Kyushu Pro Search LLP, Fukuoka, Japan), using gas chromatography–mass spectrometry as described [64].

### 4.7. 16S rRNA Sequencing and Gut Microbial Analysis

The 16S rRNA analysis was performed by a laboratory testing service (H.U. Group Research Institute G.K., Tokyo, Japan). Fresh fecal samples were collected from NDV, NDI, and HFDI mice (three mice per group), and stored at −80 °C. Fecal DNA was extracted and eluted. The microbial 16S rRNA gene was amplified (targeting the hyper-variable V3–V4 regions) as described in [65]. Sequencing was performed using Illumina MiSeq platform (Illumina, San Diego, CA, USA). The sequence data were analyzed using QIIME2 (https://qiime2.org) accessed on 21 April 2022.

### 4.8. Statistical Analyses

Data are presented as means ± standard deviation for variables with normal distribution and as medians (interquartile range) for variables with non-parametric distribution. Comparisons were made using one-way analysis of variance followed by Tukey’s post hoc test for variables with normal distribution and Kruskal–Wallis test for variables with non-parametric distribution. Values of *p* < 0.05 represent significant differences.

For gut microbial analysis, the microbial community structure was characterized using measures of α diversity and β diversity. The differences in α diversity between groups (Figure 5a) were analyzed using Kruskal–Wallis test followed by Benjamini–Hochberg test in QIIME2. The β diversity indicates differences in taxa composition between samples based on quantitative species abundance data, which are displayed as principal component analysis. Compositional similarity was compared using permutation multivariate analysis of variance followed by Benjamini–Hochberg test (Figure 5b). To identify differentially abundant taxa among groups, LEfSe analysis was performed (Figure 7) [66]. Differences between groups were considered significant when the logarithmic LDA score was >2.0 and the *p* value was <0.05.

## Figures and Tables

**Figure 1 ijms-24-14197-f001:**
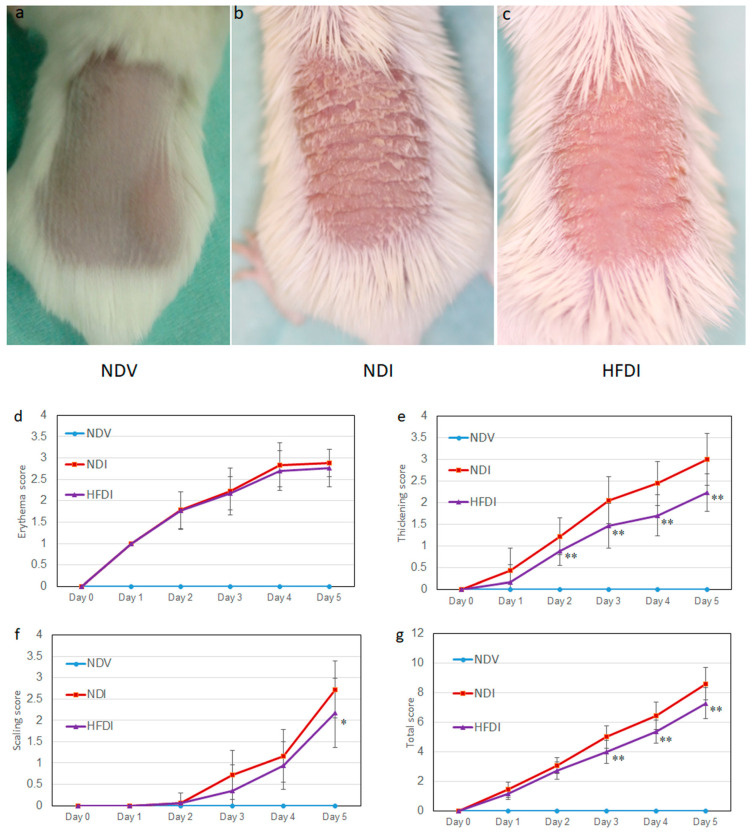
High-fiber diet (HFD) attenuates imiquimod-induced psoriasis-like dermatitis. Photos presenting shaved back skin of normal diet (ND)-fed, Vaseline-treated mice (NDV) (**a**), ND-fed, imiquimod-treated mice (NDI) (**b**), and HFD-fed, imiquimod-treated mice (HFDI) (**c**). The erythema (**d**), thickening (**e**), scaling (**f**), and total scores (**g**) are presented as means ± standard deviation (*n* = 18/group). * *p* < 0.05, ** *p* < 0.01 compared to NDI.

**Figure 2 ijms-24-14197-f002:**
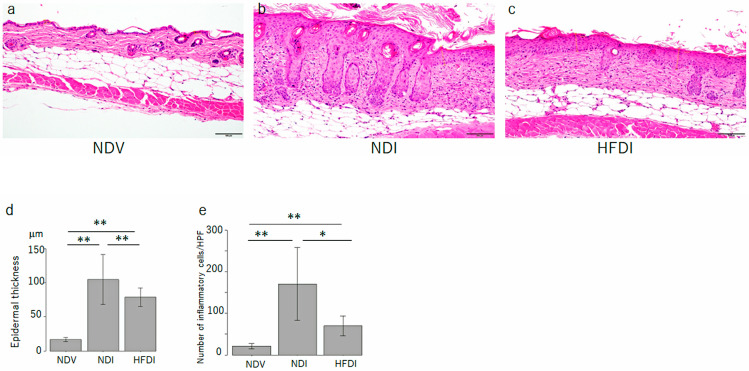
High-fiber diet (HFD) reduces acanthosis and number of inflammatory cells infiltrating in imiquimod-induced dermatitis. Hematoxylin–eosin staining skin sections from normal diet (ND)-fed, Vaseline-treated mice (NDV) (**a**), ND-fed, imiquimod-treated mice (NDI) (**b**), and HFD-fed, imiquimod-treated mice (HFDI) (**c**) (original magnification 200×). The bar in each panel represents 100 μm. The epidermal thickness (**d**) and number of infiltrating cells (**e**) are presented as means ± standard deviation (*n* = 15/group). * *p* < 0.05, ** *p* < 0.01.

**Figure 3 ijms-24-14197-f003:**
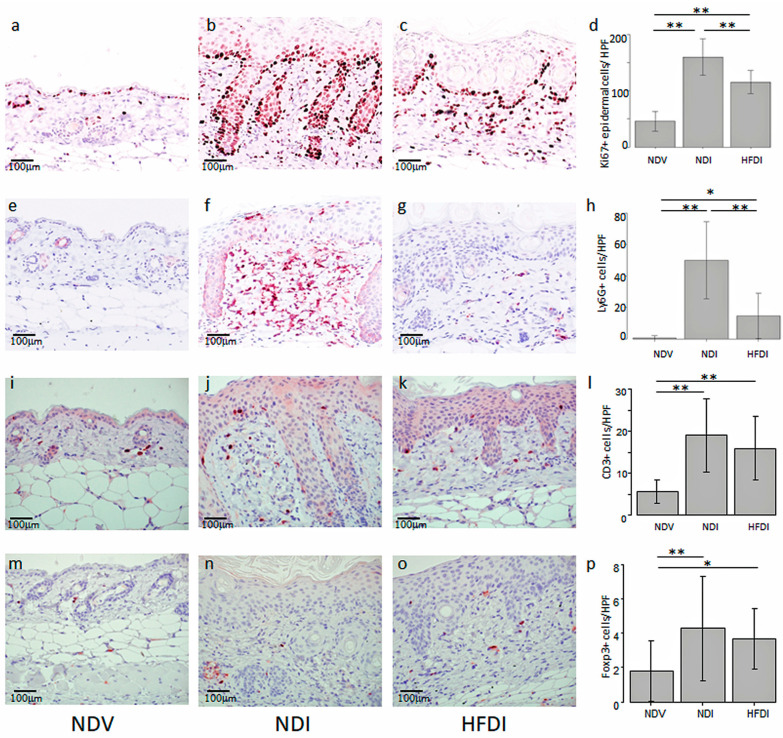
High-fiber diet (HFD) reduces the numbers of epidermal Ki67+ cells and Ly6G+ neutrophils in imiquimod-induced dermatitis. Skin sections from normal diet (ND)-fed, Vaseline-treated mice (NDV) (**a**,**e**,**i**,**m**), ND-fed, imiquimod-treated mice (NDI) (**b**,**f**,**j**,**n**), and HFD-fed, imiquimod-treated mice (HFDI) (**c**,**g**,**k**,**o**) immunohistochemically stained for Ki67 (**a**–**c**), Ly6G (**e**–**g**), CD3 (**i**–**k**), and Foxp3 (**m–o**) (original magnification 400×). The bar in each panel represents 100 μm. The numbers of cells positive for Ki67 (**d**), Ly6G (**h**), CD3 (**l**), or Foxp3 (**p**) are presented as means ± standard deviation (*n* = 15/group). * *p* < 0.05, ** *p* < 0.01.

**Figure 4 ijms-24-14197-f004:**
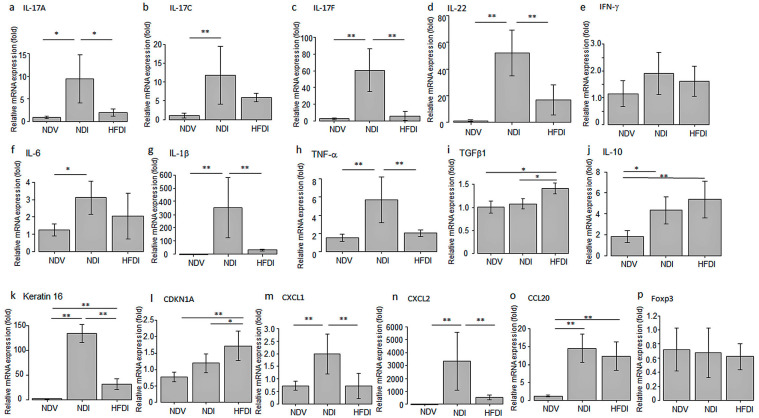
mRNA expression of IL-17A (**a**), IL-17C (**b**), IL-17F (**c**), IL-22 (**d**), IFN-γ (**e**), IL-6 (**f**), IL-1β (**g**), TNF-α (**h**), TGFβ1 (**i**), IL-10 (**j**), Keratin 16 (**k**), CDKN1A (**l**), CXCL1 (**m**), CXCL2 (**n**), CCL20 (**o**), and Foxp3 (**p**) in the skin from normal diet (ND)-fed, Vaseline-treated mice (NDV), ND-fed, imiquimod-treated mice (NDI), and high-fiber diet-fed, imiquimod-treated mice (HFDI) evaluated by qPCR. mRNA levels of individual molecules normalized to those of GAPDH are shown as fold induction relative to that of NDV, presented as means ± standard deviation (*n* = 6/group). * *p* < 0.05, ** *p* < 0.01.

**Figure 5 ijms-24-14197-f005:**
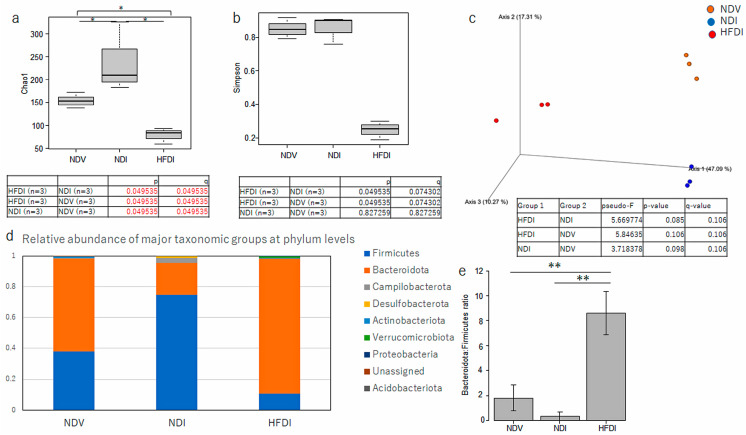
Comparison of microbial composition among normal diet (ND)-fed, Vaseline-treated mice (NDV), ND-fed, imiquimod-treated mice (NDI), and high-fiber diet-fed, imiquimod-treated mice (HFDI). The α diversity by Chao 1 index (**a**) and Simpson index (**b**) is shown as median [interquartile range] (*n* = 3/group). * *p* < 0.05 and *q* < 0.05. The red numbers in (**a**) show the statistical significance. The β diversity by Jaccard distance (**c**) is displayed as principal component analysis. Relative abundance of major taxonomic groups at phylum levels in each mice group is shown in (**d**). The ratio of relative abundance of *Bacteroidota*:*Firmicutes* is shown as means ± standard deviation (*n* = 3/group) in (**e**). ** *p* < 0.01.

**Figure 6 ijms-24-14197-f006:**
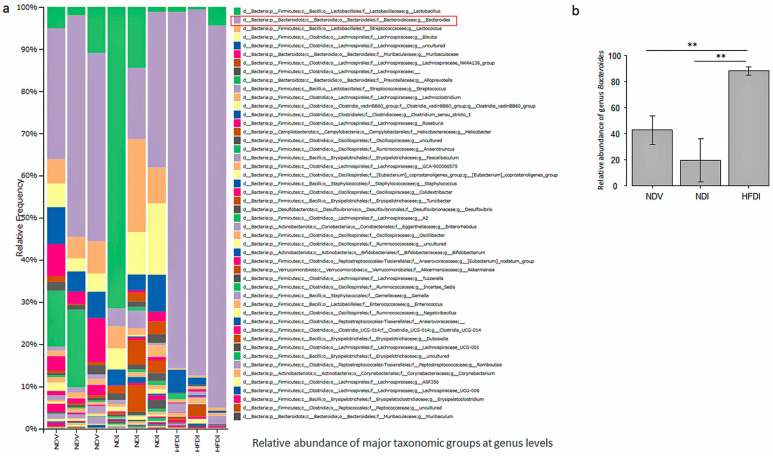
Relative abundance of major taxonomic groups at genus levels in normal diet (ND)-fed, Vaseline-treated mice (NDV), ND-fed, imiquimod-treated mice (NDI), and high-fiber diet-fed, imiquimod-treated mice (HFDI) (**a**). The genus *Bacteroides* is shown as a red rectangle. The relative abundance of genus *Bacteroides* of each mice group is presented as means ± standard deviation (*n* = 3/group) (**b**). ** *p* < 0.01.

**Figure 7 ijms-24-14197-f007:**
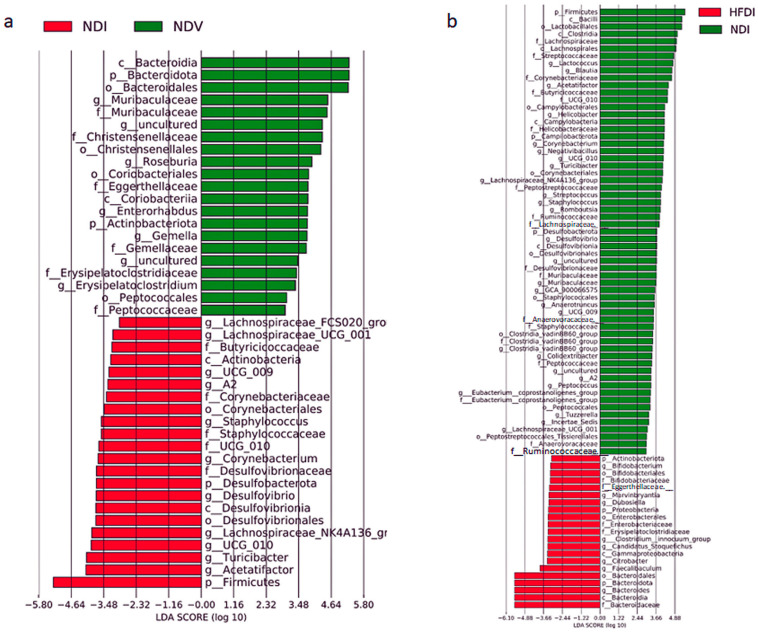
The analysis of LEfSe between normal diet (ND)-fed, Vaseline-treated mice (NDV) versus ND-fed, imiquimod-treated mice (NDI) (**a**) or between NDI versus high-fiber diet-fed, imiquimod-treated mice (HFDI) (**b**).

**Figure 8 ijms-24-14197-f008:**
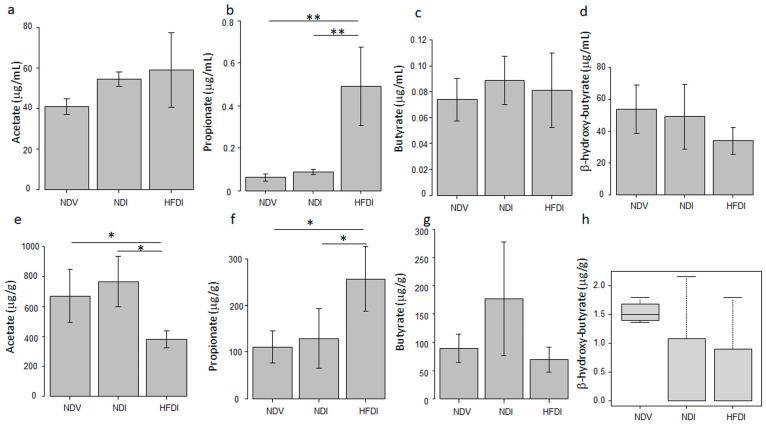
Analysis of serum (**a**–**d**) or fecal (**e**–**h**) levels of acetate (**a**,**e**), propionate (**b**,**f**), butyrate (**c**,**g**), and β-hydroxy-butyrate (**d**,**h**) in normal diet (ND)-fed, Vaseline-treated mice (NDV), ND-fed, imiquimod-treated mice (NDI), and high-fiber diet-fed, imiquimod-treated mice (HFDI). Values are shown as means ± standard deviation (*n* = 4/group) except for those of fecal β-hydroxy-butyrate (**h**) shown as median (interquartile range). * *p* < 0.05, ** *p* < 0.01.

**Figure 9 ijms-24-14197-f009:**
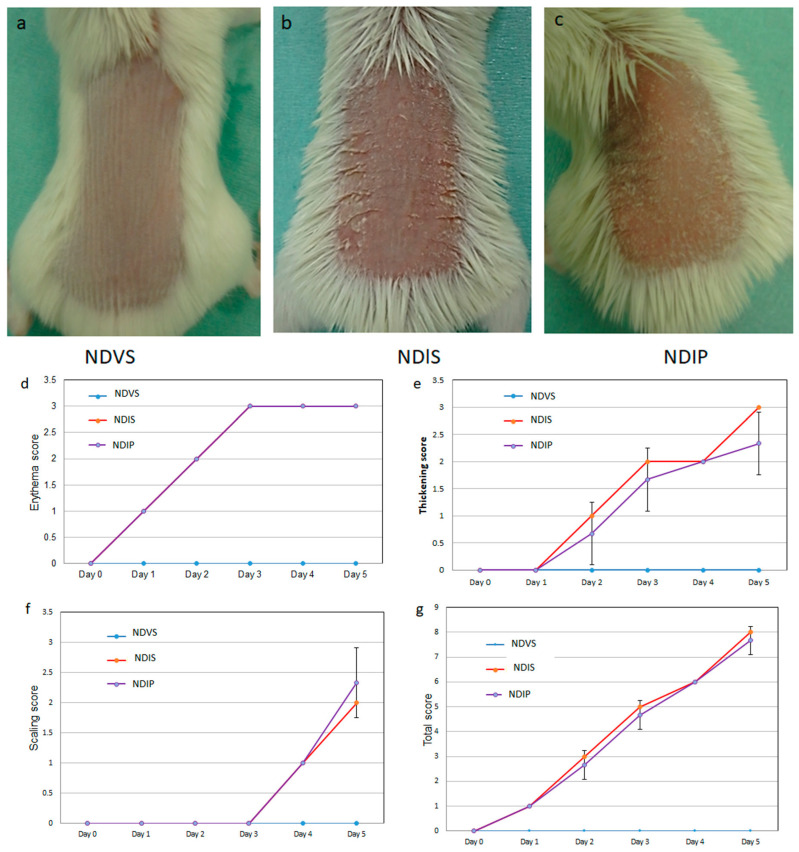
Photos of shaved back skin of normal diet (ND)-fed, Vaseline-treated, saline-drinking mice (NDVS) (**a**), ND-fed, imiquimod-treated, saline-drinking mice (NDIS) (**b**), and ND-fed, imiquimod-treated, sodium propionate-drinking mice (NDIP) (**c**). The erythema (**d**), thickening (**e**), scaling (**f**), and total scores (**g**) are presented as means ± standard deviation (*n* = 3/group).

**Figure 10 ijms-24-14197-f010:**
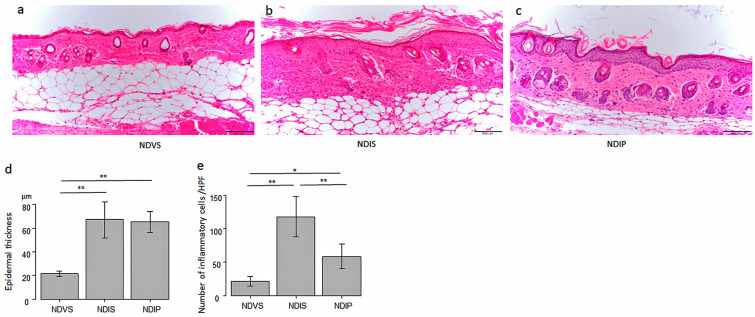
Oral propionate reduces the number of infiltrating inflammatory cells in imiquimod-induced dermatitis. Hematoxylin–eosin staining of skin sections from normal diet (ND)-fed, Vaseline-treated, saline-drinking mice (NDVS) (**a**), ND-fed, imiquimod-treated, saline-drinking mice (NDIS) (**b**), and ND-fed, imiquimod-treated, sodium propionate-drinking mice (NDIP) (**c**) (original magnification 200×). The bar in each panel represents 100 mm. The epidermal thickness (**d**) and number of infiltrating cells (**e**) are presented as means ± standard deviation (*n* = 15/group). * *p* < 0.05, ** *p* < 0.01.

**Figure 11 ijms-24-14197-f011:**
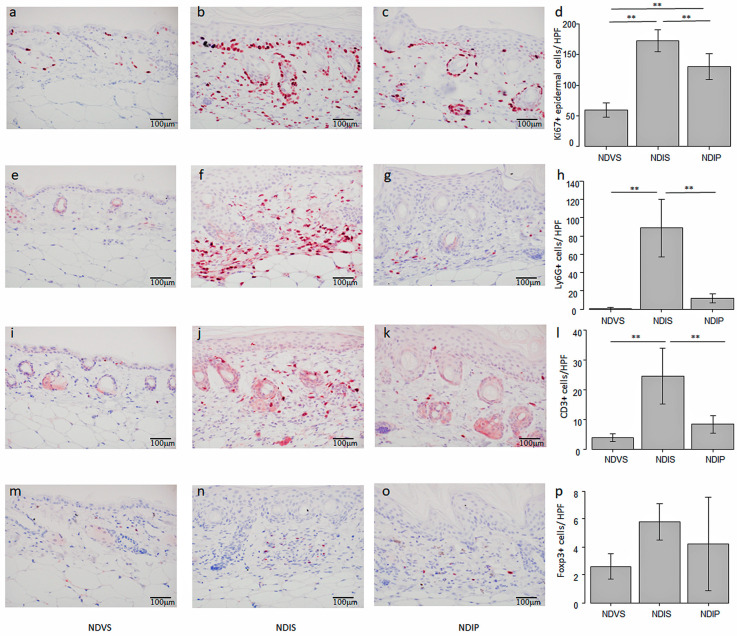
Oral propionate reduces the numbers of epidermal Ki67+ cells, infiltrating Ly6G+ cells, and CD3+ cells in imiquimod-induced dermatitis. Skin sections from normal diet (ND)-fed, Vaseline-treated, saline-drinking mice (NDVS) (**a**,**e**,**i**,**m**), ND-fed, imiquimod-treated, saline-drinking mice (NDIS) (**b**,**f**,**j**,**n**), and ND-fed, imiquimod-treated, sodium propionate-drinking mice (NDIP) (**c**,**g**,**k**,**o**) are immunohistochemically stained for Ki67 (**a**–**c**), Ly6G (**e**–**g**), CD3 (**i**–**k**), and Foxp3 (**m**–**o**) (original magnification 400×). The bar in each panel represents 100 mm. The numbers of cells positive for Ki67 (**d**), Ly6G (**h**), CD3 (**l**), or Foxp3 (**p**) are presented as means ± standard deviation (*n* = 5/group). ** *p* < 0.01.

**Figure 12 ijms-24-14197-f012:**
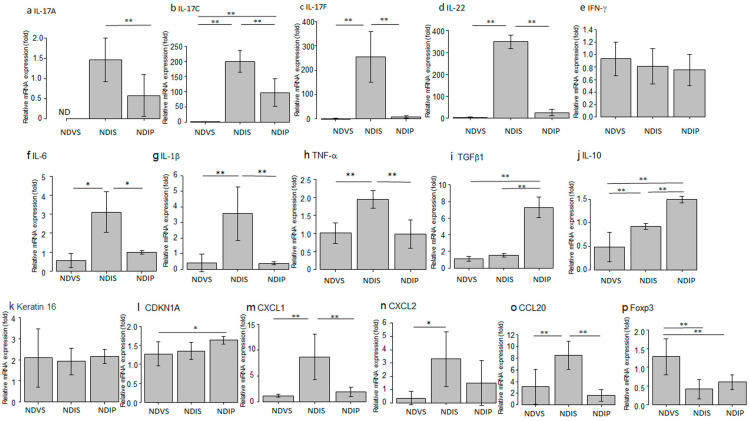
mRNA expression of IL-17A (**a**), IL-17C (**b**), IL-17F (**c**), IL-22 (**d**), IFN-γ (**e**), IL-6 (**f**), IL-1β (**g**), TNF-α (**h**), TGF-β1 (**i**), IL-10 (**j**), Keratin 16 (**k**), CDKN1A (**l**), CXCL1 (**m**), CXCL2 (**n**), CCL20 (**o**), and Foxp3 (**p**) in the skin from normal diet (ND)-fed, Vaseline-treated, saline-drinking mice (NDVS), ND-fed, imiquimod-treated, saline-drinking mice (NDIS), or ND-fed, imiquimod-treated, sodium propionate-drinking mice (NDIP), evaluated by qPCR. mRNA levels of individual molecules, normalized to those of GAPDH, are presented as fold induction relative to that of NDVS. IL-17A mRNA, normalized to that of GAPDH, was not detected in NDVS, and its levels are presented as fold induction relative to that of NDIS. Values are means ± standard deviation (*n* = 6/group). * *p* < 0.05, ** *p* < 0.01.

**Table 1 ijms-24-14197-t001:** List of agents.

List of Antibodies Used in Immunohistochemistry			
Antibody	Clone	Company	Order number
Anti-CD3	SP7	Abcam	67849
Anti-Ly6G	1A8	Aviva Systems	OATA00269
Anti-Foxp3	FJK-16s	Thermo Fisher Scientific	14-5773-82
Anti-Ki67	Polyclonal	Thermo Fisher Scientific	PA5-19462
TaqMan^®^ Gene Expression Assay IDs of genes used in quantitative real-time polymerase chain reaction			
Gene	Assay ID	Gene	Assay ID
*Ifng*	Mm01168134_m1	*Il22*	Mm00444241_m1
*Il17a*	Mm00439618_m1	*Krt16*	Mm01306670_g1
*Gapdh*	Mm99999915_g1	*Cdkn1a*	Mm04207341_m1
*Il10*	Mm01288386_m1	*Ccl20*	Mm01268754_m1
*Tgfb1*	Mm01178820_m1	*Cxcl1*	Mm04207460_m1
*Foxp3*	Mm00475162_m1	*Cxcl2*	Mm00436450_m1
*Tnfa*	Mm00443258_m1	*Il1b*	Mm00434228_m1
*Il17c*	Mm00521397_m1	*Il17f*	Mm00521423_m1
*Il6*	Mm00446190_m1		

## Data Availability

The data that support the findings of this study are available from the corresponding author upon reasonable request.

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
