# Peer review of "Dietary Fiber Inulin Improves Murine Imiquimod-Induced Psoriasis-like Dermatitis"

_ijms, 2023, doi:10.3390/ijms241814197_

Round 1
Reviewer 1 Report
The work by Yoshida and Coll. investigates the interesting relationship between diet and psoriasis. The model is appropriate and the approach complete, but I have several experimental concerns.
Major concerns:
1. Why did the Authors not present the experimental group with a normal diet without vaseline treatment? They never show that the NDV group is identical to controls, considering the potential occlusive effect of vaseline.
2. Considering the activity of imiquimod on TLR7, I trust that the investigation of the modulation of this receptor in the presented experimental conditions would be of great interest in the field
3. Why did Authors choose to perform the study on female mice? Is there any rationale? The number of animals and experiments performed must be reported in the M&M Section.
4. The description of Figure 1 in the Result section must be revised as there is no correspondence between the text and the panels.
5. How were the infiltrating cells counted (Figure 2)?
6. Although a clear decrease trend is reported for the infiltrating cells, the value in HFDI is still significantly greater than in controls
7. Considering the hypothesis of the involvement of NF-kB in oral sodium propionate-treated mice, Authors should perform at least an immunohistochemical analysis of this marker.
8. Other experimental models, mimicking the physiological environment more closely than mice, should be suggested for future studies.
Minor concerns
First of all, the Authors should describe better in the Introduction the rationale for the choice of the considered markers.
Moreover, in the revised version, the Authors must:
- include in the Figure legends only an explanation of the panels and not the methodological details
- provide the paragraph/subparagraph titles more concisely and not too detailed
- describe the experimental conditions for the immunohistochemical
- delete the last paragraph of the results (lines 307-310) as this sentence is correctly reported in the Discussion
- delete the first paragraph of the discussion as it is a mere summary of the results (lines 312-317) and highlight the more relevant and innovative findings.
- provide a clear presentation for HDAC in the Discussion
Last but not least, the hypothesis that the HFD-induced increase of systemic propionate levels derived from dietary inulin might mediate the HFD-induced attenuation of imiquimod-induced dermatitis should be suggested in the Introduction Section.
Author Response
Dear Sir or Maam,
Thank you very much for all your appreciated comments and suggestions. We did our best to improve our paper according to your remarks.
Reviewer 1
Major concerns:
1. Why did the Authors not present the experimental group with a normal diet without vaseline treatment? They never show that the NDV group is identical to controls, considering the potential occlusive effect of vaseline.
Response: In order to clarify the effects of imiquimod itself, and exclude the influence of vehicle of imiquimod cream, we have set Vaseline-treated skin, not non-treated skin, as a negative control against imiquimod-treated skin. In many papers using imiquimod-induced psoriasis model mice, not non-treated skin but Vaseline-treated skin, vehicle control, is used as negative control against imiquimod-treated skin (J Immunol 2009; 182:5836-5845; J Invest Dermatol. 2019 139(5):1110-1117; J Invest Dermatol. 2022 Aug;142(8):2194-2204.e11.). We thus followed the methods of the above papers. Vaseline-treated skin did not show abnormal findings such as erythema, scales, or skin thickness (lines 92-3) and was judged to be appropriate as a negative control against imiquimod-treated skin though we did not compare the clinical findings between non-treated skin and Vaseline-treated skin in mice.
Considering the activity of imiquimod on TLR7, I trust that the investigation of the modulation of this receptor in the presented experimental conditions would be of great interest in the field
Response: The aim of this study is to examine if inulin-supplemented diet might ameliorate the imiquimod-induced psoriasis-like dermatitis based on clinical and histological findings and expression of genes involved in its pathogenesis. The further studies might comprehensively analyze how inulin supplementation and propionate alter various signal mediators including TLR7 in the context of amelioration of imiquimod-induced psoriasis-like dermatitis. That is described in limitation section in Discussion (lines 441-3).
Why did Authors choose to perform the study on female mice? Is there any rationale?
Response: We used female mice since those were tamer and more manageable compared to male mice.
- The number of animals and experiments performed must be reported in the M&M Section.
Response: The number of mice per group is three in all the experiments except for four mice per group in SCFA measurement. The number of mice and experiments are described in Methods section (lines 482-3, 511-2, 518-9).
The description of Figure 1 in the Result section must be revised as there is no correspondence between the text and the panels.
Response: The text is revised so as that panels of figure and the text are corresponding (lines 92-96)
- How were the infiltrating cells counted (Figure 2)?
Response: The number of infiltrating inflammatory cells were measured in five random grids per section under a × 400 high-power field. That is described in Methods section (lines 498-500).
- Although a clear decrease trend is reported for the infiltrating cells, the value in HFDI is still significantly greater than in controls
Response: The results suggest that HFD may not completely reverse the inflammation induced by imiquimod, indicating that certain signals which cannot be completely counteracted by HFD may potentiate the infiltration in imiquimod-induced dermatitis.
Considering the hypothesis of the involvement of NF-kB in oral sodium propionate-treated mice, Authors should perform at least an immunohistochemical analysis of this marker.
Response: Though NF-kB is one candidate targeting signal for HFD and propionate, other signals such as STAT3, mTOR, p38 mitogen-activated protein kinase may also be the target, and comprehensive investigations for those signals involved might be performed in further studies. Since the aim of this study is to clarify if HFD may ameliorate the imiquimod-induced psoriasis-like dermatitis based on clinical and histological findings and expression of genes involved in its pathogenesis, the further detailed examinations for the signaling pathways involved are planned in future studies (lines 441-3).
- Other experimental models, mimicking the physiological environment more closely than mice, should be suggested for future studies.
Response: Other experimental models for psoriasis are suggested in Discussion; human non-lesional psoriasis skin-transplanted, global gene-manipulated, keratinocyte-specific gene-manipulated, intradermal IL-23-injected or IL-23 mini-circle DNA-delivered mice or rats, or autoantigen-recognizing T cell-transferred Rag-/- mice, or UVB photodermatitis rats are incorporated (lines 432-6). Relevant papers are newly cited (refs 22 and 61).
Minor concerns
First of all, the Authors should describe better in the Introduction the rationale for the choice of the considered markers.
Response: We have added the description for the rationale for markers investigated in immunohistochemistry and qPCR (lines 74-83). The abundant infiltration of Ly6G+neutrophils and CD3+ T cells and increase of Ki67+ proliferating keratinocytes are major histological findings in psoriasis skin lesions and imiquimod-treated skin (ref 22) while psoriasis is associated with the defect of Foxp3+ Tregs (ref 4). The expression of inflammatory or IL-17-related cytokines/chemokines, such as IL-17A/C/F, IL-22, TNF-alpha, IL-6, IL-1beta, CXCL1/2, CCL20, and of keratinocyte hyperproliferation-related molecules such as keratin 16 is enhanced while that of anti-inflammatory cytokines such as IL-10 or TGFbeta1 or of cell cycle inhibitors such as cyclin-dependent kinase inhibitor 1A is reduced in the skin lesions with psoriasis (ref 1). Thus the effects of inulin supplementation on the dysregulated expression of above markers are examined in this study.
Moreover, in the revised version, the Authors must:
- include in the Figure legends only an explanation of the panels and not the methodological details
Response: The description in the figure legends overlapping with methods section is deleted.
- provide the paragraph/subparagraph titles more concisely and not too detailed
Response: The titles are simplified (lines 89, 90, 103, 134, 160, 219, 237, 238, 284).
- describe the experimental conditions for the immunohistochemical
Response: We have detailed the conditions for the immunohistochemical analysis (lines 486-500).
- delete the last paragraph of the results (lines 307-310) as this sentence is correctly reported in the Discussion
Response: That paragraph is deleted.
- delete the first paragraph of the discussion as it is a mere summary of the results (lines 312-317) and highlight the more relevant and innovative findings.
Response: That paragraph is deleted and is displaced with one sentence showing the usefulness of HFD (line 304-5).
- provide a clear presentation for HDAC in the Discussion
Response: The explanation of HDAC and HDAC inhibitors is incorporated in the Discussion (lines 338-45).
Last but not least, the hypothesis that the HFD-induced increase of systemic propionate levels derived from dietary inulin might mediate the HFD-induced attenuation of imiquimod-induced dermatitis should be suggested in the Introduction Section.
Response: That is incorporated in Introduction (lines 83-7).
Reviewer 2 Report
The manuscript entitled ” Dietary Fibre Inulin Improves Murine Imiquimod-Induced Psoriasis-Like Dermatitis” by Mai Yoshida et al focuses on the examination if feeding with inulin-enriched high fibre diet (HFD) might improve topical imiquimod-induced psoriasis-like dermatitis in mice.
In conclusion, the authors of the article noted that dietary inulin supplementation improves imiquimod-induced psoriasis-like dermatitis partially via propionate, and may be a promising adjunctive therapy for psoriasis.
The results obtained are significant for a better understanding of the mechanism of psoriasis treatment.
The validity of the results obtained and methods suggested is unquestionable.
The following improvements in the article would help other researchers to understand the significance of the findings obtained in this work.
1. The authors should be giving the scales for Figure 3.
2. It is not clear the presentation of qPCR results when the level expression of IL17 equals 0 was analysed in parallel to GAPDH gene expression as an internal control in the figure 12.
3. The authors should give the information about the kat. numbers of antibodies used and the sequences of primers used.
Moderate editing of English language required
Author Response
Dear Sir or Maam,
Thank you very much for all your appreciated comments and suggestions. We did our best to improve our paper according to your remarks.
Reviewer 2
- The authors should be giving the scales for Figure 3.
Response: The scales are added in the figure (page 5).
- It is not clear the presentation of qPCR results when the level expression of IL17 equals 0 was analysed in parallel to GAPDH gene expression as an internal control in the figure 12.
Response: IL-17A mRNA level was analyzed in parallel to that of GAPDH as an internal control (line 300).
- The authors should give the information about the kat. numbers of antibodies used and the sequences of primers used.
Response: The catalog numbers of antibodies and the TaqMan®Gene Expression Assay IDs of genes used in this study are shown in Table1 (page 19).
Moderate editing of English language required
Response: English language is edited carefully, and several portions are improved.
Reviewer 3 Report
Thank you for your interesting study. The authors investigated the role of dietary fiber inulin in imiquimod (IMQ)-induced psoriasiform dermatitis in mice. The authors found inulin ameliorated IMQ-induced psoriasiform dermatitis. They also investigate how inulin may change microbiota in the gut. The topic is interesting and has some convining data. The animal study highlights dietary fiber inulin might be helpful for inhibiting psoriatic inflammation.
I have some suggestions for the authors.
1. The authors identified that high fiber diet increased Bacteroidota and Bacteroides. However, the findings were descriptive and the significance of these microbiota was not discussed. Please explain how these microbiota might have an impact on psoriasiform inflammation. Otherwise, the experiments of microbiota don't fit into the rationale of how inulin improves inflammation.
2. The present study investigated the role of inulin in the context of Western diet. It should be clarified that Western diet is different from high-fat diet by definition, as the sugar components are different. Literature has indicated that hyperglycemia itself is associated with psoriasiform inflammation even without high-fat (PMID: 30776434; PMID: 30571973). The reviewer suggest to mention the difference of Western diet and high-fat diet to provoke further investigations to know if inulin helps for high-fat, high-sugar Western diet or high-fat diet only or both.
Author Response
Dear Sir or Maam,
Thank you very much for all your appreciated comments and suggestions. We did our best to improve our paper according to your remarks.
Reviewer 3
I have some suggestions for the authors.
1. The authors identified that high fiber diet increased Bacteroidota and Bacteroides. However, the findings were descriptive and the significance of these microbiota was not discussed. Please explain how these microbiota might have an impact on psoriasiform inflammation. Otherwise, the experiments of microbiota don't fit into the rationale of how inulin improves inflammation.
Response: The significance of increased Bacteroidota and Bacteroides is suggested in Discussion. High fiber diet increased the relative abundance of phylum Bacteroidota and genus Bacteroides abundantly producing propionate, which may contribute to the increased systemic levels of propionate and resultant attenuation of imiquimod-induced dermatitis by propionate (lines 313-6). Further study should examine if oral gavage administration of Bacteroides may attenuate the imiquimod-induced psoriasis-like dermatitis in association with increased propionate levels (lines 438-41).
2. The present study investigated the role of inulin in the context of Western diet. It should be clarified that Western diet is different from high-fat diet by definition, as the sugar components are different. Literature has indicated that hyperglycemia itself is associated with psoriasiform inflammation even without high-fat (PMID: 30776434; PMID: 30571973). The reviewer suggest to mention the difference of Western diet and high-fat diet to provoke further investigations to know if inulin helps for high-fat, high-sugar Western diet or high-fat diet only or both.
Response: In this study, we investigated if supplementation of inulin onto normal diet, not Western diet or high-fat diet, may attenuate the imiquimod-induced psoriasiform dermatitis. However, it is reported that not high-fat and low-sugar diet but Western diet containing high-sugar and high-fat exacerbated imiquimod-induced psoriasiform dermatitis (PMID: 30571973 corresponding to ref 62) and topical imiquimod treatment induced hyperglycemia via IL-17A-mediated suppression of insulin secretion (PMID: 30776434 corresponding to ref 63). Further study should examine if inulin supplementation onto Western diet may counteract the Western diet-induced exacerbation of psoriasis-like dermatitis or imiquimod-induced hyperglycemia (lines 443-50). Relevant papers (refs 62 and 63) are newly cited.